# Interventions to Reduce Antibiotic Prescribing in LMICs: A Scoping Review of Evidence from Human and Animal Health Systems

**DOI:** 10.3390/antibiotics8010002

**Published:** 2018-12-22

**Authors:** Annie Wilkinson, Ayako Ebata, Hayley MacGregor

**Affiliations:** Institute of Development Studies, University of Sussex, Brighton BN1 NRE, UK; a.ebata@ids.ac.uk (A.E.); h.macgregor@ids.ac.uk (H.M.)

**Keywords:** antibiotic resistance, antibiotic prescribing, antibiotic use, antibiotic stewardship

## Abstract

This review identifies evidence on supply-side interventions to change the practices of antibiotic prescribers and gatekeepers in low- and middle-income countries (LMICs). A total of 102 studies met the inclusion criteria, of which 70 studies evaluated interventions and 32 provided insight into prescribing contexts. All intervention studies were from human healthcare settings, none were from animal health. Only one context study examined antibiotic use in animal health. The evidence base is uneven, with the strongest evidence on knowledge and stewardship interventions. The review found that multiplex interventions that combine different strategies to influence behaviour tend to have a higher success rate than interventions based on single strategies. Evidence on prescribing contexts highlights interacting influences including health system quality, education, perceptions of patient demand, bureaucratic processes, profit, competition, and cultures of care. Most interventions took place within one health setting. Very few studies targeted interventions across different kinds of providers and settings. Interventions in hospitals were the most commonly evaluated. There is much less evidence on private and informal private providers who play a major role in drug distribution in LMICs. There were no interventions involving drug detailers or the pharmaceutical companies despite their prominent role in the contextual studies.

## 1. Introduction

Globally, antibiotic use is increasing, up 65% between 2000 and 2015 [1]. The majority of this increase has been caused by rapid expansion of use in low- and middle-income countries (LMICs). Consumption rates in many LMICs are catching up with higher-income countries [1] but usage is often unnecessary due to lack of supportive regulation and widespread informal use [2]. While access to antibiotics is of clear public health benefit, the inappropriate use of antibiotics is a major driver of antibiotic resistance [3] and requires urgent attention. To inform context-appropriate policy in LMICs, we conducted a review of supply side interventions and their impact on unnecessary and inappropriate use of antibiotics in human and animal health systems. The aim was to determine the evidence base for changing the practices of antibiotic prescribers and gatekeepers in LMICs. 

Previous systematic reviews have found evidence to support a range of effective and safe interventions to reduce antibiotic prescribing in human health settings. Interventions that combine restrictive and enabling policies have shown better results [4]. Interventions that are interactive and multi-faceted have stronger effects, for example education with feedback and monitoring mechanisms [5,6]. However, the majority of studies included in these reviews are from high-income countries, and within that a limited range of settings (e.g., in-patient prescribing) and methods. One review [6] looked specifically for evidence on doctor’s prescribing in primary care in upper middle-income countries. However, their review only included eight studies and the authors noted that the quality of evidence was limited. We are not aware of any reviews of prescribing interventions in animal health settings. Taken together, these reviews have found encouraging results but there are questions about how generalizable they are beyond the specific targeted groups (e.g., doctors) and homogenous prescribing contexts (e.g., formal providers in relatively well organised human health systems in high-income countries). LMICs, as defined by World Bank, include a variety of health system arrangements, which makes generalizations difficult. However, in many, the human and animal healthcare contexts can be described as “pluralistic”, where people consult a variety of public, private, traditional, biomedical, formal and informal providers, and where the capacity of governments to regulate these providers is variable, and is increasingly recognised as a major concern [7]. Effective action is complicated by high rates of infection due to poor preventative public health combined with inconsistent and inequitable access to essential medicines. As such, research on LMICs interventions and contexts is urgently needed and is the focus of this review.

The monitoring of unintended consequences requires particular attention as the supply and distribution of antibiotics has the potential to include competing objectives and is sensitive to interventions not specifically targeting antibiotic use. For example, the use of rapid diagnostic tests to improve malaria diagnosis has led to an increase in antibiotic use in some settings [8,9].

The review set out to answer the following questions (for human and animal health systems):
-What supply-side interventions have been tried in LMICs to reduce inappropriate antibiotic prescribing and sale?-Where have interventions been implemented e.g., country and kind of health care settings?-Which kinds of interventions have shown promise in reducing inappropriate prescribing?-What information exists on the contexts of interventions and attempts to influence antibiotic use and prescribing?

## 2. Methods

We carried out a scoping review to provide an overview of the evidence base, as opposed to narrowing down to aggregate sub-sets of evidence as in a systematic review [10,11]. In addition to quantitative evaluations of interventions, we include qualitative studies describing the contexts of interventions to enhance understanding of the drivers of prescribing and barriers to reducing prescribing. We carried the review out in five stages: (1) identifying the research question; (2) identifying relevant studies; (3) study selection; (4) charting the data; and (5) collating, summarising and reporting the results [12]. 

### 2.1. Search Strategy

We conducted a systematic search guided by a combination of the PICO (Population, Intervention, Comparison, and Outcome) and PICo (Population/Problem, Interest, Context) approach. Full parameters are in Appendix A. PICO is usually used for quantitative reviews while PICo is used for qualitative reviews. As we were interested in both human and animal antibiotic prescribing, we looked for studies concerning people who supply antibiotics to sick people or animals, e.g., doctors and other healthcare prescribers, drug sellers, informal doctors, pharmacists, community health workers, veterinarians, farmers, and community animal health workers. We looked for any intervention aiming to influence the prescribing or sale of antibiotics (formal and informal), e.g., communication and education, stewardship programmes, treatment algorithms, delayed treatment, alternative treatments, incentives, pricing, packaging, legislation, and peer or community oversight. Outcomes of interest were measured improvement in antibiotic supply (e.g., reduction in unnecessary antibiotic prescriptions and sales, health worker adherence to guidelines, etc.), as well as reported changes in knowledge and attitudes around antibiotic prescribing, health outcomes (improved, unaffected, or adverse), and unintended consequences. Qualitative analysis of the contexts of interventions were also of interest, e.g., implementation of a new policy and descriptions of attempts to influence prescribing. 

To identify research in both the biological and social sciences, we searched MEDLINE, SCOPUS and the Cochrane database of clinical trials. We also searched the 3ie database of impact evaluations and the World Organisation for Animal Health (OIE) database. Search terms are included in Appendix B. Table 1 contains the search results for each database searched. Time and resource constraints meant we limited the date range to studies published during 2000–2017. A number of reviews already covered publications before this cut off [4,5,6], and most studies were published post-2000, so we anticipated the effect of the limitation to be minimal

### 2.2. Study Selection

Two research assistants reviewed the abstracts using the pre-defined inclusion and exclusion criteria below. Borderline studies were discussed within the team. Table 2 shows the number of studies included and excluded at each stage of the review process. From the initial figure of 15,142 search results, the review includes 102 studies. Studies of interventions and intervention contexts were separated out.

Inclusion criteria:Reports on an intervention aiming to influence the prescribing and sale of antibiotics (formal and informal), including the use of antibiotics to treat existing and suspected infections or to prevent infection (e.g., antibiotic prophylaxis)Any healthcare setting in a LMIC countryOutcomes:
Externally measured change in antibiotic use/sale/prescribingExternally measured change in knowledge and attitudes around antibiotic useSelf-reported change in antibiotic use/sale/prescribingSelf-reported change in knowledge and attitudes around antibiotic useObserved change in the quality of antibiotics prescribed/sold
Or for “context studies”
Studies that describe the contextual factors that influenced the effectiveness of interventions

Exclusion criteria:
-Studies of other ways of addressing antibiotic resistance, e.g., hygiene and infection and prevention control such as vaccines, etc.-Studies assessing adherence to antibiotics or medicine-Studies of environmental transmission of antibiotics and antibiotic resistance-Studies targeting other antimicrobials, and not targeting antibiotic use-Studies reporting on/intervening in patient demand for antibiotics and self-use-Studies evaluating effectiveness of antibiotic treatment in clinical care, e.g., comparing different treatments or ways of administering antibiotics, and reporting on clinical outcomes.-For “context studies”, those studies that only described the context and not the implementation or outcome of an intervention, e.g., a situation analysis

We carried out a detailed data extraction on the remaining 102 studies. For the 32 context studies, we extracted qualitative data summarising their findings, and indicating the setting. For the 70 intervention studies, we extracted descriptive information about the setting, intervention design and implementation, study design, the author’s assessment of impact and our own. 

We looked at intervention effectiveness in more detail in a subset of studies (*n* = 41). Studies that contained adequate information to assess effectiveness were included in this analysis. Two researchers assessed studies for inclusion. Our assessment was based on the extent to which studies contained complete information about the intervention and study design and thus whether necessary information was available and how much confidence we had in their results (e.g., sample sizes, *p*-values, primary outcomes, descriptions of the intervention, and identification of setting). Studies that comprehensively discussed their methods and outcomes, and reported strategies to minimize estimation bias and limitations were included. Studies were excluded if they were missing key study design or analysis information such as the selection and measurement of outcomes, sample sizes, or p-values. Those studies that interpreted results incorrectly or employed inconsistent methods for data collection without justification or explanation were also excluded.

## 3. Results

We present summary statistics of the intervention studies included in the review. Table 3 shows the number of studies included from each country. The largest number of studies are from China (25 out of 70 studies), followed by India (6 studies). The high number of studies in China may be in response to the major health reforms that took place there, e.g., in 2009, which saw the implementation of Essential Medicine Lists, and, during 2011–2013, when there was a national campaign on improving the use of antibacterial drugs. There are two cross-country studies—one in Brazil and Mexico, and the other in Vietnam and Thailand. The rest of the studies were conducted in a single-country setting. 

### 3.1. Intervention Settings

We found no interventions addressing antibiotic prescribing in animal health. All intervention studies were from human healthcare settings. The overwhelming majority of the reviewed studies evaluated interventions at public prescribing facilities rather than private (see Figure 1). This trend is particularly pronounced among studies conducted in primary care settings, where 15 out of 22 studies examined the public sector. The exception was for pharmacies where seven out of nine studies were conducted in the private sector. This is likely because pharmacies and drug shops are usually private enterprises. In the hospital settings, 16 and 4 studies were conducted in public and private hospitals, respectively, while in the remaining 16 it is unclear. This was due to the fact that studies mentioned hospitals’ names but we were unable to judge whether the hospitals concerned are public or private from available sources. Three other studies are not included in the figures: two were from mixed public and private sector settings, and one was not clear. 

There was also a majority of studies from urban areas, as shown in Figure 2. In total, 41 out of 70 studies looked at interventions in urban areas. In particular, studies conducted at hospitals and primary care mostly evaluated urban settings. Two of the studies carried out in mixed health care settings were from rural areas, while the remaining one study addressed both rural and urban areas. 

### 3.2. Types of Intervention 

Table 4 presents the types of intervention conducted in different settings in a “gap map” format. Interventions were classified into six broad categories: norms and standards, knowledge interventions, decision support, supply chain, economic, and organisational/management systems. Blank cells indicate that we found no evidence of an intervention being evaluated in that setting. The majority of interventions have only been studied 1–3 times, and often in only one type of setting. The evidence base is uneven, for example, there are 35 hospital-based studies and only nine in pharmacies. Only educational interventions and the essential medicine policy have been evaluated in all settings. 

Table 4 reveals a number of gaps in the evidence base though it should be noted that not all forms of intervention are as relevant for each setting. “Audit/Feedback” and “Stewardship Programme” are most often implemented in hospitals in comparison to other settings. This might reflect the ease of monitoring and management in hospital settings, which are arguably more “closed”, as well as the availability of expertise (e.g., infectious disease specialists) to perform key supportive roles. All five studies conducted on essential medicines policy in primary care setting come from China where there was major national policy reform. Although essential medicine policies are used widely, caution is needed when considering the generalisability of those studies that report on a specific policy moment and context. 

### 3.3. Intervention Impact

We examined impact in a subset of 41 studies (out of 70), selected on the basis of their reporting completeness. Studies were classified as having positive, negative, mixed, or no effect, on the basis of whether they reported improved or deteriorated prescribing rates, a combination, or no change, respectively. Studies reporting decreases in antibiotic prescribing or increased adherence to appropriate guidelines (e.g., choice of antibiotic, timing, dose, and duration of treatment) were classified as positive. Studies reporting increased antibiotic prescribing rates or decreased adherence to guidelines (e.g., increased inappropriate antibiotic use, and use of restricted antibiotics) were classified as negative. Table 5 shows the number of studies reporting each category of impact for each type of intervention. 

Twenty-one studies show positive effects of the interventions, thirteen studies document mixed effects, three studies show no effects and four studies indicate negative effects. Studies that had mixed results were those that reported both positive and negative impacts on different prescribing indicators and thus could not be categorised as wholly positive or negative. It should be noted that sometimes this may reflect more complex analysis than the studies reporting positive (or negative) results. The higher number of studies reporting positive or mixed results than negative or no results may be indicative of publication bias. Table 5 includes all intervention types identified by the review so there are some complete rows with no studies, e.g., supply chain interventions where the evidence was found and included in Table 4, but excluded here because of reporting deficiencies. 

Overall the evidence base is limited. Results are scattered and number of studies in each category of result and intervention type is small; four being the maximum number of studies reporting positive or mixed results for any intervention type. Out of the studies with positive results, the strongest evidence is clustered around knowledge interventions, and there are four studies for audit/feedback interventions and four for education/feedback interventions. Audit/feedback studies are those which introduced primarily audit interventions (e.g., reviews of prescribing patterns according to guidelines) combined with feedback mechanisms (e.g., meetings and reports). Education/feedback studies are those which introduced educational interventions (e.g., training sessions or courses and role play) combined with feedback (e.g., meetings and group discussion). There are five positive evaluations of stewardship interventions. Stewardship studies are those which combined policy review and change (e.g., new targets, development of new guidelines, and incentives/disincentives), the creation of antimicrobial/antibiotic resistance committees or working groups, education, audit and monitoring. Out of studies reporting mixed results, four studies evaluated essential medicine policies as well as public reporting (e.g., monitoring prescribing patterns and posting these publicly in/around health settings). Studies documenting no or negative effects evaluated accreditation, diagnostics, essential medicines policy, education, health insurance, and pricing intervention. More details of intervention types are in Appendix C.

Given that past reviews have found evidence suggesting that multifaceted interventions have greater impact than single interventions, we checked if similar trends were observable in our studies. We classified 21 studies as using multiple/mixed intervention strategies (e.g., education with feedback or monitoring, or stewardship programmes which are multi-strategy) and 19 as using single change pathways (e.g., one intervention modality, such as training). In one study, the approach was unclear. This study was a survey of doctor’s prescribing practice which included questions on whether they had received training on antibiotic use, however no details of the training were given. Table 6 shows that although both single and multifaceted interventions produced positive and mixed results, the majority of studies reporting mixed or positive results were multifaceted studies (positive and mixed combined = 20 versus 13) and studies evaluating single pathway interventions reported no or negative effects more frequently (negative and no effect combined = 1 versus 6). Indeed, none of the interventions classified as having mixed pathways reported negative results, and only one concluded there was no effect (see Appendix C for more detail).

### 3.4. Prescribing and Intervention Contexts

The review identified 32 context studies, from 18 different countries. One study was multi-country [13] while the rest described specific country contexts, which are listed in Appendix D. These studies described the context of interventions aiming to address antibiotic supply in hospitals, primary care and pharmacy settings, including public, private and informal providers. At least five studies looked specifically at drug shops, community pharmacists and informal village doctors. Only one study describing use of antibiotics in animal health was identified.

#### 3.4.1. Knowledge of Antimicrobial Resistance and Appropriate Antibiotic Use

A number of studies identified limited knowledge of antimicrobial resistance (AMR) and of appropriate drug use in a range of levels and sectors, for example surgeons in Jordon [14], tertiary and primary care doctors in China [15], doctors in Laos [16], public and private doctors and pharmacists in India [17] and drug sellers and pharmacists in Vietnam [18]. Bai et al. [15] found that, although age and gender did not appear to be associated with knowledge scores in Chinese settings, it was related to level of training, with more senior doctors performing well, and primary care doctors having lower levels of knowledge than tertiary care doctors. In India, urban doctors seemed to have better knowledge of AMR than rural ones [17]. Similarly, accreditation was not associated with improved knowledge and prescribing among drug sellers and pharmacists in Vietnam; however, level of training was, which in turn mapped onto urban–rural differences, with rural pharmacists being less qualified than urban ones [18]. In contrast, Reynolds and Mckee’s [19] in depth study of knowledge and practice among Chinese health workers found that awareness of resistance was high (although understanding of mechanisms was limited), but despite this health workers still prescribed inappropriately due to incentives listed in the next Section. 

#### 3.4.2. Practical Concerns and Diverse Influences on Antibiotic Prescribing

Research pointed to structural issues impeding healthcare workers’ ability to adhere to standards and guidelines. For example, in Jordan, surgeons reported that appropriate drugs were absent [14], as did pharmacists in India [20].

Beyond medicine availability, there appear to be myriad more subtle influencing factors. Studies of doctors [17,19,20] identified the following factors as influencing prescribing: (i) inadequate diagnostic facilities; (ii) lack of antibiotic guidelines; (iii) difficulty in observing patient progress (e.g., patients do not return for test results and follow up, especially in poorer areas and in public clinics where they cannot afford tests); (iv) poor intensive care facilities in rural areas; (v) patient demand for quick relief; (vi) perceived patient expectation from past prescriptions; (vii) using up stock; and (viii) fear of losing patients to competition (for private practices). Kotwani et al. [20] noted that doctors in the public sector complained they did not have enough time during each consultation to cover patient history in depth or to dissuade patients from wanting antibiotics.

Pharmacists and drug sellers [17,20,21] had a similar list of influences: (i) patient demand; (ii) a belief that cure is through antibiotics; (iii) competition from other pharmacy shops; (iv) antibiotic sales promoting business; (v) the need to use up stock; and (vi) distance and costs of reaching care facilities. Notably, public sector pharmacists in India reported overprescribing antibiotics to deal with stock which was nearing its expiry date [20]. They did not return stock as it was a lengthy process and more senior officials pressured them to use up stock. Drug shop owners in Tanzania attributed their unnecessary sales of antibiotics to: customer demand, habit (“mazoea”), following inappropriate health facility prescriptions, the need to make a profit (e.g., sell more expensive medicines), and also the need to use up stock [22]. Significantly, for these private sector providers, they were disinclined to refer patients to doctors, as it could result in their patients/customer base losing confidence in them; thus, not selling drugs had short- and potentially long-term negative effects on profit.

An enlightening study from Vietnam [18] found that antibiotics were not the most profitable drug for sellers, although it was the most commonly sold drug and the mark up did vary considerably; antibiotics contributed 24% and 18% of total revenue in urban and rural pharmacies, respectively, while herbal remedies contributed 24% and 27%. 

#### 3.4.3. Industry Influence

The influence of the pharmaceutical industry in providing information and incentives looms large [17,23,24,25]. In Nigeria, both public and private hospitals’ doctors indicated that industry was their primary source of information about drugs [23]. Indian doctors and pharmacists reported coming under pressure and being offered incentives by pharmaceutical companies to use or sell drugs, especially newer brands [17]. A study of doctors in Pakistan found that private sector doctors were (reportedly) more susceptible to patient demand and to fear of losing patients to other providers than their public sector counterparts, while public sector doctors were more likely to face problems with the availability of drugs [26]. In Laos, doctors in provincial settings were more likely to receive information from drug companies than doctors in district and central hospitals [16]. 

In their study of healthcare worker incentives in China, Reynolds and Mckee [19] found that some doctors had arrangements with pharmaceutical companies to split the profits of extra sales. This was in addition to the sanctioned cost recovery incentives (pre-2009 Chinese policy reforms). 

#### 3.4.4. Care and Responsibility

Studies identified additional cultures of care that influenced prescribing, suggesting the idea that prescribers and sellers are primarily influenced by stock and profits may be an over simplification. Indian pharmacists said the antibiotics they sold to poor patients was akin to social work [20]; and Indian doctors said they prescribed antibiotics as prevention as their patients living with poor sanitation and in unhygienic conditions were susceptible to infection [27]. There was also some reluctance for pharmacists to challenge doctor’s authority by contesting prescriptions even if they knew they were unnecessary [20]. 

#### 3.4.5. Perceptions of Public Perception

Many of the studies include health worker’s perceptions of the public and their demands. A common perception among doctors is that patients do not know about diseases and desire antibiotics as a result [17]. In Nigeria [23], doctors say that when the public pay for health services, i.e., in private hospitals, they expect medicines to be in their bills. Providers in both the public and private sector stated that patient demand was a key factor in their prescribing [15,17,19,20,22,26,27,28]. However, it is not clear what evidence these views are based on and whether they involve misperceptions on the part of providers. In Sri Lanka, Tillekeratne et al. [28] found that although patients admitted expecting drugs they were not specifically demanding antibiotics. 

#### 3.4.6. Perspectives on Interventions 

Some studies asked participants for suggestions of interventions to improve antibiotic use, which included: creating public awareness, better healthcare provider communication, improved diagnostic support, stricter implementation of guidelines, continuing education, and strengthening of regulations [17]. There was an appetite for training [16,29]. For example, in Laos, a large majority of doctors surveyed were in favour of educational programmes, whereas only 45% of doctors thought guidelines were effective [16]; indeed, some doctors thought that guidelines were more of an obstacle than a help (22%). South African medical students expressed a preference for e-learning platforms, but significantly did not appreciate more interactive learning techniques such as “problem-based learning” or “registrar interaction” (i.e., interaction with senior colleagues, in this case trainee specialists) [30]. This could reflect poor implementation of these learning modalities but it suggests that clarification is needed of their role in education. 

#### 3.4.7. Policy Coordination 

In Bangladesh, the progressive and once highly acclaimed National Drug Policy of 1982 has not been maintained [31]; the regulatory institutions do not have power to curb prices or excess use-generics less available, high mark up and sale of antibiotics for cough and fever. Bangladesh’s experience serves as an important reminder, for example in relation to China’s apparently effective essential drug reform, that there must be sustained investment into the comprehensive implementation of policy especially those which address antibiotic use as part of other agenda’s such as universal health coverage. A policy analysis in Namibia [32] highlighted inconsistency in national policy frameworks. While treatment guidelines (e.g., HIV, malaria, tuberculosis, etc.) included recommendations about antibiotics there was no overarching guideline for antibiotics; meanwhile, many of the antibiotics included on the essential medicine list were not included in the treatment guidelines, and vice versa. The study noted that lack of coordination between the committees developing these lists had led to confusion with implications for clinical practice. Similarly, in China, doctors complained about the number of new guidelines and no single authoritative source [19]. It is relevant that the only international study, a survey of hospital-based stewardship initiatives [13], found that major barriers included funding and human resources, suggesting sustainable implementation is a challenge. 

#### 3.4.8. Unintended Consequences

The management of fevers and common illnesses in outpatient settings is a significant problem. A study examining the appropriateness of treatment in Papua New Guinea [33] found there was poor compliance with the Integrated Management for Childhood Illness guidelines (ICMI) and that 40% of children were not treated accurately: 29% received antibiotics when they should not have and 11% did not receive antibiotics when they should have done. They observed that the prescription of antibiotics was strongly influenced by the result of rapid diagnostic tests (RDTs) for malaria in feverish patients: in children presenting with criteria for mild pneumonia (who should receive antibiotics), only 40% were treated when the malaria RDT was positive, compared to 76% when the test was negative. This suggests that health workers are primarily following RDT results and ignoring other clinical symptoms, meaning the management of co-morbid bacterial and malarial infections are poorly managed. Several studies point to the inadequacies of apparently simple methods (e.g., fever charts or RDTs) of managing fevers and antibiotic use successfully at the community level [34,35]. Comprehensive follow up of patients who may have co-morbid infections or who may have been misdiagnosed should be a priority. 

#### 3.4.9. Animal Health

Finally, only one context study covered use of antibiotics in agriculture. Aquaculture farmers in Vietnam were asked about AMR [36]. Approximately half of farmers surveyed used antibiotics, primarily as prophylaxis, which they bought directly from manufacturers not through veterinarians. Farmers were not aware of regulations about antibiotic use and their main sources of information was from drug manufactures, including in seminars jointly arranged by the companies and local government. 

## 4. Discussion

The review confirms previous studies [4,5,6] that have found that multiplex interventions combining different strategies to influence behaviour tend to have a higher success rate than interventions based on single strategies. Another similarity to other reviews [4] is that many of the interventions which worked well combined restrictive and enabling strategies, i.e., educational techniques combined with forms of monitoring. To the best of the authors’ knowledge, this is the largest review to document these patterns in LMIC countries. 

The inclusion of evidence on prescribing contexts provides important additional insight. These studies highlighted a web of interacting influences on prescriber behaviour that cannot be reduced to simple motivations, such as profit. Instead, health system quality and availability, education, perceptions of patient demand, bureaucratic processes, competition, and cultures of care all play a role. 

Physician perceptions of patient demand [15,17,19,20,22,26,27,28] requires more attention as observations do not always support this [28]. Indeed, elsewhere studies that have observed clinical interactions have shown how doctors prescribe irrespective of patient demand [37] and that patient satisfaction is not necessarily linked to whether a prescription was received [38], although these studies are old. A more recent study from multiple European settings [39] found that, while patients frequently hope for and expect antibiotics, it is less common for them to ask for them explicitly. Patients were prescribed antibiotics considerably more often than they asked for them (54% compared to 10%, respectively), and more than was clinically necessary. The same study found that, in general, patients were satisfied with their consultations, whether prescribed antibiotics or not. Assessing the level and influence of patient demand is difficult as research is often based on reported behaviour and perceptions rather than observations.

It is striking the extent to which different providers blame others within the system: health workers blame patients, pharmacists blame doctors and their incorrect prescriptions, and drug sellers blame competitors saying patients will just “go somewhere else”. Such blame shifting highlights a more general importance of interactions between people within the health system: pharmacists fear challenging doctors’ prescriptions, drug sellers fear losing their customers, and many doctors and pharmacists get a large proportion of their information about antibiotics from drug detailers who also offer incentives. In discussions of interventions to address AMR, the temptation is to focus on awareness raising and changing the behaviour of individuals, be they healthcare workers including prescribers and pharmacists, or patients. However, the stewardship, supply and use of antibiotics in human and animal healthcare is best viewed with a broad systemic lens [2,40]. This encompasses an appreciation of how the behaviour of individuals is mediated by a range of structural and contextual factors operating at different scales, which can affect individual agency and decision-making. 

Scholars have advocated for a consideration of behavioural change interventions within the context of everyday social practices [41]; for a whole system approach (at macro, meso and micro levels) to analysing policy outcomes in the health sector [42]; for a consideration of formal and informal health markets for drugs and services for both people and animals [7,43]; for an appreciation of incentive structures in drug supply chains and the effects on over-the-counter drug retailers and informal providers [44]; and for an approach to healthcare worker behavioural change interventions that appreciates individual rationales as well as the contextual settings in which specific behaviours are practiced [45]. The overall view is also one of decentred governance and regulation, which departs significantly from traditional “command and control” models [46]. It is notable therefore that while a majority of studies we identified in this review implemented multiplex interventions, most of these took place within one health setting, e.g., a stewardship programme in a hospital. Very few studies targeted different kinds of healthcare provider and different kinds of health settings. One exception was by Hoa et al [47] who attempted an educational programme for all providers in one district, public, private, formal and informal and primary or secondary care. In this example, however, there was a particularly high drop-out rate among informal providers illustrating the challenges of engaging hard to reach groups in interventions. Moreover, important interactions between types of providers are not being addressed and it is rare to find examples of whole system approaches which have been recommended [2,40]. China provides one of the few examples of system-wide interventions at the national and regional level. There is a need for coordinated action and research across multiple settings and actors throughout and between the human and animal supply chain. 

Significantly, our study was unable to identify a single study on interventions that addressed prescribing behaviour for veterinary use of antibiotics, and only one context study on agricultural use. Evidence is emerging that antibiotics critical for human health are being used in animal farming [48] including colistin [49,50]. Research needs to address agricultural use of antibiotics and its overlap with human health system in terms of the antibiotics used, healthcare personnel, and transmission pathways of resistant pathogens. Measures to improve essential medicine use provides a precedent for success [51]. The National Action Plan process provides an opportunity to facilitate this. In particular, employing the “One Health” lens [52,53], taking the intersection among human, veterinary, and environmental health into account, can help address the multi-sector nature of AMR and develop systematic strategies to tackle this challenge. 

In addition to the major lack of evidence documenting the effectiveness of interventions to improve veterinary prescribing, this review has identified some gaps in the human evidence. Studies in hospitals were most common, possibly as hospitals are more observable and controllable environments. The overwhelming majority of studies evaluated interventions at public facilities. There was also a majority of studies from urban areas, with less from rural areas, possibly because hospitals are over represented in the data and tend to be in urban areas. Private providers, and in particular those in primary care settings, is another gap. Informal private providers are especially underrepresented. Evidence from LMICs suggests that rural and/or poor people extensively rely on healthcare services provided by informal practitioners and suppliers of drugs [54,55]. Therefore, understanding the antibiotic prescribing practices of the informal providers, and improving them, has important implications for health of many marginalized people. There were no interventions in this review involving drug retailers or the pharmaceutical companies despite their prominent role in the contextual studies. 

Public understanding and demand is a critical piece of the puzzle and requires action but it must be based on accurate understandings of lay knowledge and demand otherwise messaging and educational interventions will fail. Research is needed into public views and their influence on demand and prescriber behaviour. This should form part of a holistic view of supply and demand. 

Our study has a number of limitations. While the literature search was extensive, it may not be exhaustive and studies prior to 2000 were not included. The large number of heterogeneous studies meant we were unable to assess quality of the evidence in detail and we did not compare or combine the effects of selected studies. Further research is needed to assess the strength and magnitude of effects for promising interventions. A majority of studies (*n* = 20) reported positive effects while far fewer reported mixed, negative or no effects (*n* = 13, 4 and 3 respectively). This indicates that publication bias may be an issue. Nevertheless, the review was able to identify trends and gaps in the evidence base. 

## 5. Conclusions

The evidence collected in this review comes from a range of health care settings, for example hospitals, primary care, pharmacists or drug shops, and from interventions targeting different types of health providers including doctors, nurses, and drug sellers. This review has identified evidence on interventions to improve antibiotic use among providers in LMIC settings, which, in the authors’ opinion, has not been well represented in previous reviews. The review found that multiplex interventions that combine different strategies to influence behaviour tend to have a higher success rate than interventions based on single strategies. However, the evidence base is uneven with hospital and urban contexts over-represented for interventions. There is much less evidence on private providers, especially in primary care settings. Informal private providers who play a major role in drug distribution in LMICs are especially underrepresented. Furthermore, there were no interventions involving drug detailers or the pharmaceutical companies despite their prominent role in the contextual studies. Strikingly, no study was identified that addressed veterinary prescribing of antibiotics. 

Evidence on prescribing contexts highlights a web of interacting influences on prescriber behaviour including health system quality and availability, education, perceptions of patient demand, bureaucratic processes, profit, competition, and cultures of care. These contextual studies underscore the importance of interactions between different people within the health system. Although a majority of studies implemented multiplex interventions, most of these took place within one health setting, e.g., a stewardship programme in a hospital. Very few studies targeted different kinds of health provider and interactions across different kinds of health setting. There is an urgent need for coordinated multi-actor studies including multiple settings, by taking a One Health approach, including agricultural settings, and actors throughout the supply chain. 

## Figures and Tables

**Figure 1 antibiotics-08-00002-f001:**
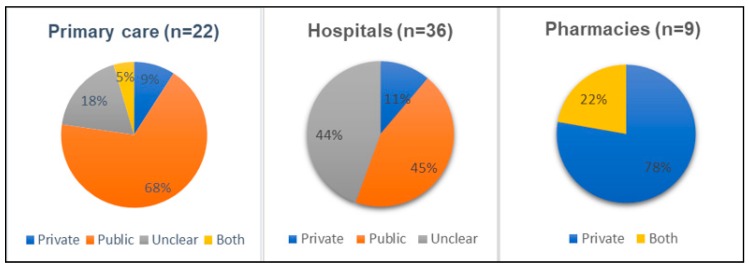
Sector where intervention was tested, by type of health setting.

**Figure 2 antibiotics-08-00002-f002:**
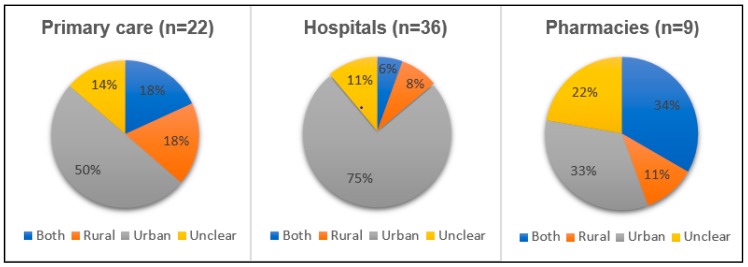
Location (urban or rural) where intervention was tried, by type of health setting.

**Table 1 antibiotics-08-00002-t001:** Search results.

Database	Result
Scopus	20,083
Cochrane Central Register of Controlled Trials (CENTRAL)	3823
3ie Impact Evaluations	26
World Organisation for Animal Health (OIE)	0
Total	32,066
Total after duplicates removed	17,716
Total 2000–2017	15,142

**Table 2 antibiotics-08-00002-t002:** Selection process.

Stages	
Abstract review	15,142
Excluded (did not meet inclusion criteria)	14,734
Included-	408
Full text review	408
Excluded—did not meet inclusion criteria on review	289
Duplicates (e.g., studies reporting same data)	4
Full text not available/not in English	13
Included—context studies	32
Included—intervention studies	70
Total—context and intervention	102

**Table 3 antibiotics-08-00002-t003:** Country settings.

Country, by World Health Organisation Region	
African region	15
Kenya	3
Mali	1
Malawi	1
Nigeria	3
South Africa	2
Sudan	2
Tanzania	3
Region of the Americas	5
Argentina	1
Brazil	2
Brazil and Mexico	1
French Guiana	1
European region	7
Republic of Srpska, Bosnia and Herzegovina	1
Serbia	2
Turkey	4
Eastern Mediterranean region	3
Pakistan	1
Iran	2
South-East Asian region	13
India	6
Indonesia	1
Nepal	3
Thailand	2
Thailand and Vietnam	1
Western Pacific Region	27
China	25
Vietnam	2
Total	70

**Table 4 antibiotics-08-00002-t004:** Kinds of intervention, by setting.

Type of Intervention	Hospital	Primary Care	Pharmacies	Mixed Settings	Total
Norms and standards (formal and informal)					
Accreditation			1		1
Guidelines	3				3
Public reporting		4			4
Restrict over-the-counter sales			2		2
Prescription control	1				1
Knowledge					
Audit/Feedback	8	2			10
Education	2	3	2	2	9
Education/Feedback	3	2			5
Education/Feedback/Regulation			2		2
Information	1				1
Decision support					
Algorithms	1	2			3
Diagnostics	2			1	3
Supply chain					
Decentralisation of Supply		1			1
Drug Delivery	1				1
Economic					
Financial incentives		1			1
Pricing strategy		1			1
Health Insurance	2	1			3
Organisational/management systems					
Essential Medicine Policy	2	5	1		8
Stewardship Programme	10		1		11
Total	36	22	9	3	70

Number of studies: 
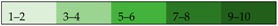
.

**Table 5 antibiotics-08-00002-t005:** Summary of intervention impacts.

Type of Intervention	Reported Impact				Total
	Positive	Mixed	Negative	No Effect	
Norms and standards (formal and informal)					
Accreditation				1	1
Guidelines	2				2
Public reporting		4			4
Restrict over-the-counter sales		2			2
Prescription control					0
Knowledge					
Audit/Feedback	4				4
Education	2			1	4
Education/Feedback	4	1			4
Education/Feedback/Regulation	1				1
Information					0
Decision support					
Algorithms	1	1			2
Diagnostics	1		1		2
Supply chain					
Decentralisation of Supply					0
Drug Delivery					0
Economic					
Financial incentives	1				1
Pricing strategy			1		1
Health Insurance			1		1
Organisational/management systems					
Essential Medicine Policy		4	1	1	6
Stewardship Programme	5	1			6
Total	21	13	4	3	41

Number of studies 
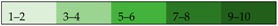
.

**Table 6 antibiotics-08-00002-t006:** Results, by intervention approach.

Reported Results	Intervention Pathway		Total
	Single	Mixed/Multifaceted	
Positive	8	12	20
Mixed	5	8	13
Negative	4	0	4
No effect	2	1	3
Total	19	21	40

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
