# Peer review of "Interventions to Reduce Antibiotic Prescribing in LMICs: A Scoping Review of Evidence from Human and Animal Health Systems"

_antibiotics, 2018, doi:10.3390/antibiotics8010002_

Round 1

Reviewer 1 Report

Overall, this is a robust, comprehensively reported scoping review, which provides a valuable addition to the existing knowledge base around prescribing interventions in LMICs. The inclusion of both intervention and context-based studies is a strength. Table 4 is particularly interesting in terms of identifying current gaps in the evidence. The interpretation and discussion of results is insightful.

Addressing the following recommendations would strengthen the report:

1.       It would be helpful if an operational definition of low-middle income countries (LMICs) was offered in the introduction.

2.       A justification for limiting the search to articles published after 2000 should be provided – the strongest reviews do not necessarily limit the search based on age of publication; where this occurs, a rationale should be provided.

3.       Line 133 – 143 it is not clear why a ‘subset’ of studies were examined in more detail; is this perhaps more about the completeness of information in each report to enable further analysis, rather than the identification of a subset per se. The phrasing of this paragraph implies that there was a selection of studies included for more in-depth review, rather than, where this information was reported, it was reviewed. This paragraph could be re-phrased for greater clarity.

4.       In table 5 & associated narrative (line 213), a definition of ‘stewardship programmes’ would be helpful i.e. what did these consist of that differentiated them from other interventions e.g. education +audit & feedback?

Author Response

Thank you for your positive comments. We have addressed all your comments and present the details of the changes we made below.

1.       It would be helpful if an operational definition of low-middle income countries (LMICs) was offered in the introduction.

We have used the list published by World Bank and specified this on page 2.

2.       A justification for limiting the search to articles published after 2000 should be provided – the strongest reviews do not necessarily limit the search based on age of publication; where this occurs, a rationale should be provided.

This was largely due to time and resource constraints and we explain this on page 3 between lines 96-99. We also explain that this cut off is unlikely to have resulted in bias as there have been a number of review articles covering the period before 2000 that can be consulted. The total number of articles published concerning antibiotic prescribing has increased substantially since 2000, a further reason for this choice.

3.       Line 133 – 143 it is not clear why a ‘subset’ of studies were examined in more detail; is this perhaps more about the completeness of information in each report to enable further analysis, rather than the identification of a subset per se. The phrasing of this paragraph implies that there was a selection of studies included for more in-depth review, rather than, where this information was reported, it was reviewed. This paragraph could be re-phrased for greater clarity.

Thank you for this comment. We have explained more details on page 4. We selected a subset of articles because these 41 articles demonstrated clarity and scientific rigour whereas others had missing information.

4.       In table 5 & associated narrative (line 213), a definition of ‘stewardship programmes’ would be helpful i.e. what did these consist of that differentiated them from other interventions e.g. education +audit & feedback?

We clarified this on page 8 where we explain that stewardship refers to programmes that combined policy change, the creation of AMR committees or working groups, education, audit and monitoring.  We also added a sentence to direct readers to Appendix C where we provide more details

Thank you for your time in reviewing our paper. 

Reviewer 2 Report

A really valuable manuscript that adds to the body of evidence. 

Well written

Some statements need to be 'toned down' as authors cannot definitely confirm these esp as many studies in LMIC are difficult to find and not always published in indexed journals 

Some specific comments 

133: please specify/quantify what subset is 

196-197 - I think should be made clearer what positive, negative refers to.Eg if assuming respectively, then positive = increased prescribing which in AMR context would be inappropriate 

"Studies were classified as having positive, negative, mixed, or no effect, on the basis of whether they reported increases or decreases in prescribing rates, a combination, or no change. "

Within  table will recommend stating reported impact of intervention rather than results 

209: Stewardship had 6 positive or mixed 

237: ? studies missing from first part of sentence? 

242: define what more specialist means 

303: a word appears to be missing after nigeria 

311: McNulty C M (PHE) has studied doctors perceptions of patient demand. These are more recent and should be included alongside Scandinavian studies. They also show that patients are generally satisfied as long as they feel heard, that they didn't waste theirs and doctors time. This is the evidence behind the non prescription pads in the uk 

317: ? More effective than....

320: registrar interaction is not clear. Pls clarify. Either because that was the group studied or it actually refers to registrants? Also is interaction registrar one level down from consultants (in UK but us this the same in all other countries) training more junior doctors? 

355-369 appears to be in the wrong section. And earlier it was stated no studies in animal health. But farmers using antibiotics as prophylaxis is animal health, so whilst the earlier statements were about interventions, it is worth clarifying there is a study on agriculture? 

365 is a bold statement that could be challenged, suggest c,arifying that authors are aware of. Many such reviews are often published in journals not easily accessible 

378 - suggest healthcare providers changed to healthcare professionals or perhaps workers  including prescribers and pharmacists? I didn't note any studies focused on nurses. Also healthcare providers tend to be the organisation not the individuals

Use of healthcare providers in 394 seems more appropriate 

414: interventions isn't the only gap, there isn't even a similar study to eg agriculture 

424 -clarify  no interventions  found in this review 

442-443 - in the opinion of the authors. Not fact 

458 - add agriculture and perhaps one health approach in brackets 

Not sure if I missed it but refer to appendix(supplementary info)  within text where relevant  

Author Response

Thank you for your comments and suggestions. We have incorporated all of them in the following way.

1)      Some statements need to be 'toned down' as authors cannot definitely confirm these esp as many studies in LMIC are difficult to find and not always published in indexed journals 

We have done so throughout the text. Some specific comments 

2)      133: please specify/quantify what subset is 

The subset of studies add up to 41. We have indicated this on page 4 and explained that we limited our analysis of impact a subset of articles which demonstrated clarity and scientific rigour whereas others had missing information.

3)      196-197 - I think should be made clearer what positive, negative refers to.Eg if assuming respectively, then positive = increased prescribing which in AMR context would be inappropriate "Studies were classified as having positive, negative, mixed, or no effect, on the basis of whether they reported increases or decreases in prescribing rates, a combination, or no change. "

Thank you for this comment. We have clarified this on page 7. We refer to positive results as studies reporting decreases in antibiotic prescribing or increased adherence to guidelines. Negative results refer to those reporting increased antibiotic prescribing rates or decreased adherence to guidelines.

4)      Within table will recommend stating reported impact of intervention rather than results 

We have changed the wording as recommended.

5)      209: Stewardship had 6 positive or mixed

We are not clear what edit is required here. Table 5 shows that stewardship programmes had 5 studies with positive result and 1 study with mixed result, which matches what is reported in the narrative.

6)      237: ? studies missing from first part of sentence? 

We have corrected this sentence.

7)      242: define what more specialist means 

We have modified this sentence and explained that more senior doctors performed well, and also that primary care doctors had lower levels of knowledge than tertiary care doctors.

8)      303: a word appears to be missing after nigeria 

We have corrected this.

9)      311: McNulty C M (PHE) has studied doctors’ perceptions of patient demand. These are more recent and should be included alongside Scandinavian studies. They also show that patients are generally satisfied as long as they feel heard, that they didn't waste theirs and doctors time. This is the evidence behind the non prescription pads in the uk 

Thank you for pointing this out. We were not able to find the McNulty study, at least not a paper which refers specifically to doctor’s perceptions of patient demand. However, we have now included a more recent study by Coenen et al which reports on a large cross country European study and compares doctor’s accounts with those of patients. It seems that many of these studies are based on reported behaviour which is a weakness. We have also moved these references and the related discussion from the results section into the ‘discussion’ section as we felt it was more appropriate there. See lines 400-408.

10)   317: ? More effective than....

We have rephrased this sentence.

11)   320: registrar interaction is not clear. Pls clarify. Either because that was the group studied or it actually refers to registrants? Also is interaction registrar one level down from consultants (in UK but us this the same in all other countries) training more junior doctors? 

Sorry this was not clear. ‘Registrar interaction’ was taken from the original paper and refers to student interaction with more senior colleagues, in this case with registrars. The paper is from South Africa where ‘registrar’ means trainee specialist. We have clarified this now.

12)   355-369 appears to be in the wrong section. And earlier it was stated no studies in animal health. But farmers using antibiotics as prophylaxis is animal health, so whilst the earlier statements were about interventions, it is worth clarifying there is a study on agriculture? 

We added a section on animal health e.g. section 3.4.9. The study on animal health we present is one of the context studies. We have added a sentence to clarify this in the abstract, and at the beginning of  section 3.4.

13)   365 is a bold statement that could be challenged, suggest c,arifying that authors are aware of. Many such reviews are often published in journals not easily accessible 

We have clarified that this is to the best of our knowledge.

14)   378 - suggest healthcare providers changed to healthcare professionals or perhaps workers  including prescribers and pharmacists? I didn't note any studies focused on nurses. Also healthcare providers tend to be the organisation not the individuals

We have changed the wording.  

15)   Use of healthcare providers in 394 seems more appropriate 

We have changed the wording here, too.  

16)   414: interventions isn't the only gap, there isn't even a similar study to eg agriculture 

We have changed the sentence to “In addition to the major lack of evidence documenting the effectiveness of interventions to improve veterinary prescribing, this review has identified some gaps in the human evidence.”

17)   424 -clarify  no interventions  found in this review 

We have clarified this on page 13.

18)   442-443 - in the opinion of the authors. Not fact 

We have added “in the authors’ opinion”.

19)   458 - add agriculture and perhaps one health approach in brackets 

We changed “veterinary” to “agricultural” and added “by taking a One Health approach” in the sentence.

20)   Not sure if I missed it but refer to appendix(supplementary info)  within text where relevant

We have done this throughout the text

Thank you for you time reviewing our paper.